## PERSPECTIVE

# Increasing the reproducibility of fluid biomarker studies in neurodegenerative studies

Niklas Mattsson-Carlgren [1,2,3✉], Sebastian Palmqvist[1,4], Kaj Blennow [5,6] & Oskar Hansson [1,4✉]

Biomarkers have revolutionized scientific research on neurodegenerative diseases, in particular Alzheimer's disease, transformed drug trial design, and are also increasingly improving patient management in clinical practice. A few key cerebrospinal fluid biomarkers have been robustly associated with neurodegenerative diseases. Several novel biomarkers are very promising, especially blood-based markers. However, many biomarker findings have had low reproducibility despite initial promising results. In this perspective, we identify possible sources for low reproducibility of studies on fluid biomarkers for neurodegenerative diseases, with a focus on Alzheimer's disease. We suggest guidelines for researchers and journal editors, with the aim to improve reproducibility of findings.

Neurodegenerative diseases, including Alzheimer's disease (AD), account for significant morbidity, mortality, and costs worldwide. A major problem for research, clinical practice, and drug development is that diagnosis, prognosis and disease monitoring are difficult using clinical examination alone. Clinical examination is particularly problematic in early disease stages and cannot often on its own guide diagnosis or predict progression. Biomarkers have been introduced as a way to improve this, by providing objective measures of the underlying pathophysiology[1]. Very successful results have been achieved for fluid biomarkers, including both in cerebrospinal fluid (CSF) and blood[2]. For AD, biomarkers of relevant brain changes have even been incorporated into research definition of the disease (using biomarkers for β-amyloid [Aβ] pathology, tau pathology, and neurodegeneration)[3]. Examples of highly reproducible biomarkers that to some degree are used in clinical practice for diagnosis of neurodegenerative diseases include CSF Aβ42, the Aβ42/40 ratio, total-tau (T-tau) and phosphorylated tau (P-tau) for AD diagnosis[4], and real-time quaking-induced conversion (RT-QuIC) assays on CSF for CJD[5–7]. CSF levels of neurofilament light (NFL) protein[8] is also sometimes used in clinical practice as a disease-unspecific biomarker for neuronal injury, to detect the presence of degeneration and thereby support a diagnosis of a neurodegenerative condition, e.g., Amyotrophic Lateral Sclerosis (ALS)[9]. When disease-modifying treatments for common neurodegenerative diseases become available, biomarkers may also have a role to guide usage of treatments in clinical practice. This may be relevant very soon, giving the promising recent results of certain immunotherapies against AD[10–12].

However, for many very promising biomarker findings, including biomarker panels[13–16], replication efforts have failed[17]. Poor reproducibility is, however, not a problem isolated to

[1] Clinical Memory Research Unit, Faculty of Medicine, Lund University, Lund, Sweden. [2] Department of Neurology, Skåne University Hospital, Lund University, Lund, Sweden. [3] Wallenberg Center for Molecular Medicine, Lund University, Lund, Sweden. [4] Memory Clinic, Skåne University Hospital, Malmö, Sweden. [5] Clinical Neurochemistry Laboratory, Sahlgrenska University Hospital, Mölndal, Sweden. [6] Institute of Neuroscience and Physiology, University of Gothenburg, Gothenburg, Sweden. ✉email: niklas.mattsson@med.lu.se; oskar.hansson@med.lu.se

biomarker research[18,19]. In 2016, the journal *Nature* published results from a survey taken by 1576 researchers from many scientific disciplines, where 52% thought that there was a "significant crisis" in reproducibility of published research, due to factors such as selective reporting, pressure to publish, low statistical power or poor analysis, too little replication in the original lab, publication bias, and other factors[20]. The problem is aggravated by the fact that small studies that overestimate effects (including for biomarker performance) may be more likely to get cited than larger studies with more sobering results[21]. Publication of biomarker findings with low reproducibility is a waste of time and money for researchers and assay developers aiming to replicate the results.

In this perspective article, we therefore analyze reproducibility issues for fluid biomarkers for neurodegenerative diseases. This is not a systematic quantitative review of all fluid biomarker studies that have been published for neurodegenerative diseases. We have selected examples of studies that represent different types of reproducibility problems. Most of our examples are from the AD-field, but we believe that the recommendations are applicable as guiding principles also for other neurodegenerative diseases. We explore several amendable sources of poor reproducibility, including cohort design, pre-analytical and analytical factors, and statistical procedures. We suggest factors that may be taken into consideration when designing and publishing biomarker studies. Our ambition is to help researchers, drug developers and scientific journals to achieve reproducible results and advance the field of biomarkers in neurodegenerative diseases.

## Factors affecting reproducibility

There are many potential sources of poor reproducibility for biomarker studies. These include cohort-related factors, pre-analytical factors, analytical and kit-related factors, biotemporal variability of the measured markers, insufficient statistical methods, lack of proper validation, and factors related to the decisions to submit or accept publications for publishing. Several of these factors contribute to poor reproducibility by either introducing imprecision (a random error) of measurements (increasing "variability"—the closeness of agreement between biomarker measurements obtained by replicate measurements on the same or similar objects under specified condition[22]), or introducing bias (a systematic error) of measurements (reducing "accuracy"—the closeness of agreement between a biomarker measurement and the true value of the biomarker[22]). We summarize examples of these factors and considerations in two figures. The first part of studies, from cohort recruitment up until biomarker measurements (including cohort-related, assay-related, pre-analytical and analytical factors) are summarized in Fig. 1. The later part of studies and validation efforts (including choices of relevant comparisons, statistical analyses, and different levels of validation) are summarized in Fig. 2. Each of these components are described in detail below.

### Cohort-related factors

Suboptimal cohort design increases the likelihood of overoptimistic findings. First, studies with small patient and control samples typically overestimate performance compared to larger studies[23,24]. This is logical, since only large effects can be detected with a small sample. If by chance a larger effect is present in the smaller sample this may increase the likelihood of publication, overestimating the effect in an initial small pilot study and leading to publication bias. A funnel plot may indicate if publication bias is present across a number of studies[25].

Second, if study participants are pre-selected, or if extensive inclusion or exclusion criteria are applied, findings may have lower reproducibility compared to if participants are consecutively or randomly recruited. This is because in a more heterogenous (more real-world like) sample, several factors may contribute to both the biomarker and the clinical endpoint, attenuating the biomarker effect. For biomarkers developed to detect and quantify a specific pathology, such as amyloid deposition, it is important to have a control group not harboring such brain changes (e.g., having negative amyloid positron emission tomography (PET) scans or CSF Aβ42/40 ratio). However, one should typically avoid to select a "super healthy" control population that not only differs substantially from subjects with neurodegenerative diseases but also from the general population without such diseases, in order to avoid bias of the results.

Third, study procedures should not differ between patients and controls. If patients and controls are recruited at different centers or during different time periods, known or unknown systematic differences in the procedures may lead to a bias in form of biomarker findings that are erroneously interpreted as disease-related. Optimally all groups are recruited at the same sites, during the same period using the same standardized operating procedures.

Fourth, confounding or modifying factors may impact reproducibility. This may include demographics, genetic factors, drugs, kidney and liver function, or presence of co-morbidities. For example, in highly educated patients, "cognitive reserve" may attenuate the relationship between a biomarker and clinical diagnosis. Many late-onset clinically diagnosed AD cases also have combinations of α-synuclein pathology, TDP-43 deposits, microvascular changes and hippocampal sclerosis on top of plaque and tangle pathology[26], which may attenuate associations between biomarkers and the studied disease (since different pathologies may affect biomarkers in different ways, e.g., white matter lesions may reduce CSF levels of several biomarkers[27]).

Lastly, we think that it is important to pre-register the cohort study (e.g., at clinicaltrials.gov), both to increase transparency of the biomarker study, and to determine outcomes before data collection and analysis, to reduce selective reporting and p-hacking.

### Assay-related factors

Analytical methods (assays) can be subject to both random and systematic measurement errors, which can all impact reproducibility of findings. The procedures to control assay-related factors are complex and technical, but necessary for the introduction of assays in routine diagnostic use in Clinical Chemistry. We will only briefly highlight a few aspects here.

For ligand binding assays (e.g., immunoassays), two key properties are specificity and selectivity[28]. The assay specificity refers to how well the assay (with its antibodies and other components) can distinguish between its intended analyte and other structurally similar components. Poor specificity leads to a systematic overestimation of biomarker levels. Assay specificity can be tested by evaluating the assay for cross-reactivity using similar material, e.g., a protein homolog such as Aβ40 in an Aβ42 assay or the medium or heavy subunits of neurofilament in an NFL assay, which it should not react towards. The assay selectivity refers to how well the assay measures the analyte in the sample matrix, with presence of other biological components. Selectivity can be tested with different spike-recovery experiments, where a known quantity of the measured analyte is added to a sample, and the assays' ability to recover the known quantity is evaluated. One caveat is that for protein biomarkers, the spiking material is often recombinant and may differ from the endogenous form of the biomarker, e.g., by lacking post-translational modifications, truncations etc. Hypothetically, this

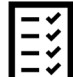 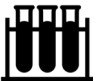

**I. Cohort-related factors**

1. Minimize inclusion and exclusion criteria
2. Adhere to a random or consecutive recruitment
3. Apply similar procedures to both control and patient groups
4. Ensure that the study is powered to detect the expected effect

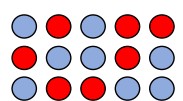 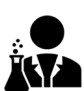

**II. Assay-related factors**

1. Optimize assay specificity (distinguish the analyte from similar components) and selectivity (quantification in sample matrix)
2. Handle interferents, dilution linearity, parallelism etc
3. Implement certified reference procedures and materials
4. Monitor and minimize lot-to-lot variability

**III. Preanalytical factors**

1. Establish and follow standardized operating procedures
2. Standardize sampling time point and methodology (e.g. equipment and tubes), sample handling (e.g. pipetting, centrifugation, aliquotation), and sample storage procedures
3. Be watchful of unknown sources of variability and bias!

**IV. Analytical factors**

1. Follow manual (kit inserts)
2. Keep laboratory technicians blinded to clinical data
3. Analyze samples in a randomized fashion
4. Include internal and external quality control samples
5. Regularly maintain and calibrate instruments/equipment

**Fig. 1 Finding reproducible biomarkers, from cohort recruitment to biomarker measurements.** A flowchart highlighting key points described in the review, with examples of key factors that influence reproducibility from the first design of the study, up to biomarker measurements.

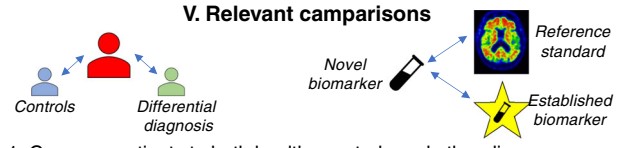

**V. Relevant camparisons**

1. Compare patients to both healthy controls and other diseases
2. If applicable, test separate stages or forms of the disease
3. Compare to neuropathology and/or an independent reference standard
4. Compare performance to previously established biomarkers with similar utility and applicability

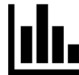

**VI. Statistical analyses**

1. Test several models, e.g. for cross-sectional diagnosis, prediction of future progression, and correlations between longitudinal biomarker changes and other measures
2. Correct for multiple comparisons when needed
3. Perform internal cross-validation (test set, training set)
4. Report both overall measures of diagnostic accuracy (e.g. area under the receiver operating characteristic curve), and sensitivity and specificity at relevant cut-offs

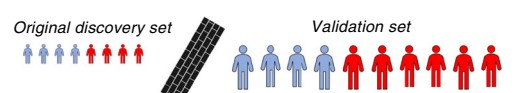

**VII. Initial indenpendent validation**

1. Adhere to the original preanalytical and analytical protocols
2. Use an independent validation cohort (w/o overlapping biases)
3. Increase the heterogeneity of participants (e.g. primary care)
4. Apply and validate previously defined cut-points and models (instead of establishing new cutpoints or models)

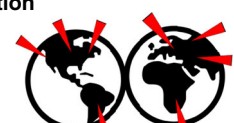

**VIII. Global validation**

1. Replication by other research groups
2. Analyze multiple batches/ kit lots
3. Test effects of different preanalytical factors
4. Harmonize and compare different assays
5. Establish certified reference procedures and materials
6. Establish universal protocols and cut-points
7. Publish negative as well as positive results

**Fig. 2 Finding reproducible biomarkers, from relevant comparisons to validation.** A flowchart highlighting key points described in the review, with examples of key factors that influence reproducibility from including relevant comparisons to global validation.

may over- or underestimate the accuracy of the assay compared to when measuring the biomarker in its endogenous form.

Other key properties that needs to be controlled for in assay development and maintenance include dilution linearity (measurement levels should be proportional to sample dilution), and parallelism (standard reference and serially diluted sample curves should be parallel)[29]. In individual samples, assays may be sensitive to interferents such as lipids, hemolysate or heterophilic antibodies, which may result in both over- or underestimation of the measured concentrations of the analyte[30].

Assay-related factors are not only important to control when developing and launching a new assay, but also over time when using an assay in research or clinical practice. As known or unknown changes occur in production procedures or reagents, biomarker measurements may be affected by lot-to-lot variability. This can introduce differences both between studies, and within studies (e.g., if several lots of analytical kits are used within a large study). It is the responsibility of kit manufacturers and vendors to minimize lot-to-lot variability. Batch-bridging at individual labs can track this potential issue[31], which can be controlled for by rejection of a batch or possibly by adjustments of calibrator levels.

If possible, novel assays should always be compared with certified reference procedures, using certified reference materials[32–34]. Unfortunately, reference measurement procedures ("gold standard" methods) or certified reference materials ("gold

standard" samples) are not available for most neurodegeneration biomarkers, with the exception of CSF Aβ42[35].

**Pre-analytical factors**. Even with a validated assay, many factors can affect biomarker measurements even before the analytical phase starts. Examples of these include time-of-day for sampling, the technique used by the doctor/nurse/staff for phlebotomy or lumbar puncture, tube-related factors, and pure errors in tube handling and labeling[36].

Some biomarkers may vary due to normal physiological processes, or in response to diet, stress factors, or health issues. For example, plasma T-tau (but not P-tau) may be affected by sleep loss[37] (which might contribute to the poor reproducibility of plasma T-tau as a biomarker, described below). Studies of test-retest variability with repeated measures (after hours, days or weeks) may quantify physiological, changes of the biomarker over time. One study tested variability over eight weeks for CSF Aβ38, Aβ40, Aβ42, T-tau, NFL, and panels of inflammatory, neurovascular, and metabolic biomarkers[38]. The variability was acceptable for most of biomarkers, but a few inflammatory markers showed instability over time, making them less suitable as CSF biomarkers (SAA, CRP, and IL-10). Some biomarkers may also have a circadian, cyclic, variability, where levels fluctuate over the course of a 24-hour cycle. This is difficult to study for CSF biomarkers, since the sampling procedure itself, with repeated CSF collection through an indwelling lumbar catheter, affects CSF turn-over and biomarker concentrations much more clearly than diurnal changes[39]. To minimize the influence of potential biotemporal variability, it is recommended that sampling is done at a consistent time of the day. However, one study even suggested that some biomarkers (CSF Aβ40, Aβ42, T-tau, and P-tau were tested) may fluctuate over the year[40], which (if reproduced) could be an additional source of variability.

Beyond factors that temporarily offset biomarker levels, some biomarkers may have a large normal physiological range within the population, which could impact the likelihood that a biomarker reaches a threshold for positivity. For example, a person with a slightly higher than normal release of Aβ peptides into CSF is more likely to be classified as normal for CSF Aβ42 compared to a person with slightly lower than normal release of Aβ peptides. Most of the Aβ peptides released into CSF are of the shorter variant, ending at the amino acid position 40 (Aβ40), which is not affected by AD. The associations for CSF Aβ42 with AD are, therefore, improved when adjusting for CSF Aβ40, usually in the form of the CSF Aβ42/Aβ40 ratio[41], which adjusts for between-individual differences in overall (with Aβ40 serving as a proxy for total) Aβ peptide levels in CSF, and potentially also for both within-individual and between-individual differences in production and clearance of CSF[42]. For this reason, we include CSF Aβ42/Aβ40 rather than just CSF Aβ42 in the clinical work-up of dementia patients at our centers. The usage of reference peptides may also improve the accuracy for other AD biomarkers in CSF and blood, but more work is needed in this area.

Beyond these biological factors, sampling and storage procedures may also affect biomarker levels, and thereby influence reproducibility for both blood-based and CSF-based biomarker[43,44]. For example, CSF Aβ42 is hydrophobic and especially sensitive for variations in pre-analytical handling, and may be affected by the type of tubes used for CSF sampling and storage, as well as freeze-thaw procedures[45]. As another example of a pre-analytical storage effect, a 12.5 kDa C-terminally cleaved fragment of cystatin C was proposed as a promising CSF biomarker for multiple sclerosis (MS), reaching 100% specificity for other neurological disorders[46]. However, a paper by another research group could not replicate the finding[47] and it also was

noted that this protein cleavage occurs as a storage artifact when samples are being stored at −20 °C instead of −80 °C (which had been shown before as well[48]).

To minimize all pre-analytical factors, it is important that the pre-analytical protocol is identical for patient and control groups, and across participating centers in multi-center studies. Further, it is important that systematic experiments are performed to evaluate the effects of different pre-analytical variables on the biomarker of interest, even though such publication seldom are published in higher impact journals. See e.g., ref. [9] for a standardized protocol applied in clinical practice to improve the performance of CSF Aβ42.

**Analytical factors**. Several analytical factors (beyond what we describe above under assay-related factors) may increase imprecision or bias and thereby contribute to poor reproducibility of studies.

Broadly, analytical imprecision (variability) can be described in terms of within-lab and between-lab variability. Within-lab variability can be further divided into intra-assay variability (precision in the same run, which can be monitored by measuring duplicates of all or selected samples), and inter-assay variability (precision across different assay runs, which can be monitored by running aliquots of the same internal control samples in every run). To minimize these sources of variability, laboratory staff must follow analytical protocols carefully, and have sufficient control systems to detect and quantify variability. Significant between-lab, inter-assay, or lot-to-lot variability makes it difficult to reproduce results at specific cut-points or decision thresholds, and is a hurdle towards widespread use of the biomarker[49]. Significant variability also makes it difficult to use longitudinal rates of change as indicators of incipient pathology. Longitudinal testing of biomarkers with low variability may provide meaningful information, even before critical thresholds are reached, as shown for Aβ PET imaging[50]. An international quality control program has been established to monitor measurement variability for different assays and platforms for key CSF biomarkers for AD[49,51]. This program has demonstrated that variability can be reduced by transferring assays from manual to fully automated methods, as shown for the CSF AD biomarkers Aβ42[52], T-tau and P-tau[53] (see updated info at http://www.neurochem.gu.se/theAlzAssQCprogram).

To minimize analytical bias, technicians running the assays must be blinded to all clinical data. One consequence of this is that all samples become randomized, to make sure that patients and controls are spread out over the plates. In the best scenario the whole lab analyzing the samples should not have access to any clinical information until the data is finalized.

**Statistical methods associated with poor reproducibility**. A typical situation with a high risk for false-positive findings is when a large number of biomarkers are tested without a grounded theory. This is especially common in "omics"-studies (e.g., proteomics or metabolomics) with hundreds or thousands of molecules. A statistical adjustment for multiple comparisons is always required at some step for these studies, but if two cohorts are available, we can accept false-positive findings in the discovery cohort and apply a strict adjustment when testing the biomarker in the validation cohort. We also note the risk for overcorrection (leading to "type II error") when several findings have nominal uncorrected significance close to an a priori threshold, and they are all ruled out as non-significant after correction for multiple comparisons.

A special scenario with high risk for poor reproducibility is when multiple biomarkers are tested simultaneously in a panel

selected from a large set of originally available biomarkers. Such panels form complex models, which are liable to overfitting (thus representing noise in the data rather true patterns of interest). Again the most robust strategy to avoid false-positive findings is to use a large external validation cohort to test the specific biomarker panel identified in the discovery cohort. When only one cohort is available, validation methods should be applied within the single cohort, as explained below.

To elucidate if biomarker changes are specific to a certain neurodegenerative disease or if alterations are non-specific in response to brain injury, we suggest that studies include comparisons between as many different diagnostic groups as possible. For example, a study of a novel AD biomarker should optimally compare Aβ-positive AD dementia, other neurodegenerative diseases, Aβ-positive MCI, Aβ-negative MCI, Aβ-positive cognitively unimpaired controls, and Aβ-negative cognitively unimpaired controls[54]. If the biomarker is specific for AD, we expect alterations both in Aβ-positive AD dementia and MCI (and potentially also in Aβ-positive cognitively unimpaired controls if changes come early). If the biomarker is altered also in other neurodegenerative diseases and Aβ-negative MCI and dementia, it indicates that it responds non-specifically to brain injury. If a biomarker is altered only in Aβ-positive MCI but not in AD dementia, we consider it likely to be a false-positive finding (although a transient peak in biomarker levels in earlier stages is possible and has been suggested[55], our experience is that this has not been reproduced for a biomarker of neurodegeneration).

We also suggest studies to not only report a measure of overall performance to separate between groups, e.g., AUC, which may give a "numerical" impression of high performance, since it cannot be lower than 0.5, but also sensitivities and specificities (and possibly positive and negative predictive values depending on the generalizability of the disease prevalence in the present cohort) preferably using cut-offs defined in another population. This is since sensitivity and specificity figures are important for the clinical application, either identifying as many as possible with a disease (high sensitivity) but accepting false positives, or ruling out a disease (high specificity). It is also valuable to show cut-points defined using different methods, including for example at optimal specificity (such as the mean plus two standard deviations in Aβ-negative cognitively unimpaired individuals for an AD-biomarker), a combined optimal sensitivity and specificity (Youden index), or a "natural" cut-points identified by Gaussian mixture models, which can provide robust cut-points especially for biomarkers with a bimodal distribution[56–58]. For clinical chemistry tests used in clinical practice, reference intervals are most often based on findings in healthy individuals, and for biomarkers where changes in both directions are clinically relevant (such as plasma glucose) the cut-offs corresponds to the central 95% of the distribution, or for biomarkers where only a change in one direction (either an increase or a decrease) is clinically relevant (such as increased serum Troponin-T) the single cut-off corresponds to the upper 95% of the distribution[59].

**Validation methods**. All studies with novel or unexpected biomarker results should have a validation section. The strongest possible validation is to demonstrate robustness of results in a separate validation cohort. Robustness should be shown both for the overall continuous associations between the biomarker and the main clinical endpoints, and if possible for performance at specific cut-points (e.g., for classification). The validation cohort must be sufficiently large to be powered to detect the biomarker effect found in the discovery cohort.

If an independent validation cohort is not available, validation is often done within the original cohort[60]. One possibility is to split the available sample into a training set (e.g., 80%, but may be higher if the sample size is small) where the biomarker is "trained" to predict the outcome, and a test set (e.g., 20%), for validation. The partitioning of the cohort should be done before any analyses, to avoid leakage of information from the test to the training set. If researchers first find effects for a biomarker in the whole population, and then post hoc perform a training/test split, the risk is high for overoptimistic estimates of performance. A preferable alternative to a training/test split is to perform $k$-fold cross-validation (CV). The data is portioned into $k$ bins of equal size (usually $k = 10$, but may be lower with a small sample size). In an iterative procedure, the biomarker model (for example a logistic regression model for binary classification) is trained sequentially in all bins expect one, and evaluated in the remaining single bin. The result is a string of $k$ measures of performance. This is preferable to a simple training/test set split, because it reduces the impact of the random grouping of subjects into one of the sets, and gives a distribution of the test effect, rather than just one test performance value. The robustness of the analysis can be further increased by repeating the $k$-fold CV (e.g., a five times iterated 10-fold CV). Confounding factors can be balanced across bins. Some statistical models include "hyperparameters" (e.g., the regularization constant in LASSO regression, or the number of clusters in k-means clustering). Hyperparameters should be estimated separately from the model performance. For this, nested CV may be used, with two layers of CV ("inner" and "outer"), where the hyperparameters are tuned in one layer, and the model performance is estimated in the second layer. However, if the discovery and validation cohorts are based just on dividing the total cohort into two, possible systematic bias between patients and controls in terms of, e.g., differences in pre-analytical procedures or cohort specific biases will remain. Consequently, there are previous examples of convincing internal cross-validations that have failed when replicated in an independent cohort[15,17]. Therefore, the discovery and validation cohorts need to be independent and come from different studies.

One type of desirable (but rarely available) validation, is towards neuropathological data[61]. This is valuable since clinical diagnosis is imperfect, with about 70% sensitivity and specificity[62]. However, neuropathological validation also has its caveats. First, the size of cohorts with autopsy data are often small. Second, there is also almost always a lag between fluid biomarker sampling and death, which may underestimate associations between biomarker levels and brain changes that likely continue to develop, or other types of pathology (e.g., ischemic lesions) may appear. Third, most neuropathological examinations does not include detailed quantification of brain changes across the whole brain, but is focused on particular tissue sections, which may be more or less representative of the pathologies that produce altered biomarker levels. Moreover validation against neuropathology most often restricts the analysis to generating data on biomarker performance in the very latest stages of the disease. A proxy for pathological validation of Aβ and tau pathology may in some cases be molecular imaging, using well-validated PET tracers[63].

Finally, validation of a biomarker with a secondary, independent assay, increases the chance that the biomarker signal is true, i.e., that the assay actually measures what it is intended to measure. Correlations between different assays, or assay formats, are typically evaluated in Round Robin or Commutability studies, in which aliquots of the same samples are analyzed using several different analytical methods[32]. There are several examples of biomarkers that correlate poorly when measured with different assays[17], making links to the underlying biological processes unclear. Associations should also be tested towards previously validated biomarkers of pathology (including for example CSF Aβ42/Aβ40 ratio or Aβ PET imaging for Aβ pathology).

Researchers and Editors could work together to define community standards for the type of validation that is necessary for biomarkers detected through unbiased methods. Completely novel exploratory markers should have extensive validation (including validation in an independent cohort, which should be large enough to have sufficient power). For more established biomarkers, an internal validation procedure may be sufficient. Complex models, especially including panels of biomarkers, or biomarkers used together with demographic factors, need more extensive validation than a single a priori defined biomarker. Single biomarkers selected post hoc from a panel of many exploratory biomarkers needs careful validation, specifically that the exact composition of the panel is set in the discovery cohort, and optimally that the performance of the same panel is evaluated in a fully independent validation cohort. Extensive validation is necessary if the cohort is at risk for bias. Finally, to our knowledge, almost all biomarkers that have been reproduced sufficiently to be used in clinical practice (e.g., CSF Aβ42 and P-tau for AD, and RT-QuIC for CJD) have been discovered with a grounded theory, based on a clear hypothesis about disease mechanisms. Despite decades of research, unbiased methods, including proteomics and metabolomics have still not resulted in biomarkers that have been sufficiently reproduced for use in clinical practice. We therefore argue that biomarkers detected through unbiased methods should have extensive validation before publication.

**Levels of reproducibility**. We have reviewed the literature for reproducibility of fluid biomarkers in neurodegenerative diseases, including AD, Parkinson's disease (PD) and related conditions, frontotemporal lobe dementia, and motor neuron disease. In summary, we found that a few biomarkers have had very high reproducibility with almost unanimously converging results. These biomarkers are summarized below ("Rank I"). There is a second large group of biomarkers with variable results, which may be considered to have uncertain reproducibility. We present a few examples of such biomarkers, with suggested explanations for the variable results ("Rank II"). Finally, for many biomarkers replication has been attempted and failed. Again, we present a few examples, together with a discussion about the reasons for the failed replications ("Rank III"). These are summarized in Fig. 3.

**Rank I: high reproducibility**. A few fluid biomarkers have been robustly associated with neurodegenerative diseases. These biomarkers are often incorporated in clinical trials, either at study inclusion to enrich for participants with AD pathology, or as exploratory secondary outcomes. Some of these biomarkers are also used in clinical practice for dementia work-up.

CSF Aβ42, T-tau, and P-tau are altered in AD, as reviewed extensively before (e.g., in the database AlzBiomarker[25,64]). CSF T-tau has been tested in AD dementia versus controls in at least 188 studies, including over 12000 patients and over 8000 controls. All but two of these found significantly higher CSF T-tau in AD, while the two negative studies found non-significant increases. However, CSF T-tau is also increased non-specifically due to brain injury in other neurological diseases. CSF P-tau (mainly using the phosphorylation variant P-tau181) has been tested in AD dementia versus controls in at least 116 studies. All but three of these found significantly higher CSF P-tau in AD (the three negative studies found non-significant increases). CSF Aβ42 has been tested in AD dementia versus controls in at least 168 studies. All but seven of these found reduced levels in AD; and only one of those seven studies found a significant increase in AD (see https://www.alzforum.org/alzbiomarker). Several studies have also found that CSF Aβ42, T-tau, and P-tau are altered prior to dementia in

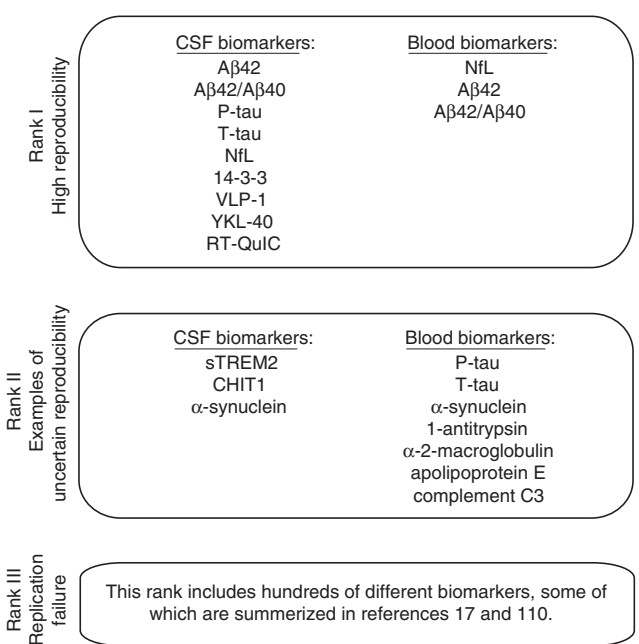

**Fig. 3 Ranks of reproducibility.** A brief summary with examples of fluid biomarkers for neurodegenerative diseases, by their level of reproducibility.

AD, demonstrating that the biomarker changes are robust at all clinically relevant stages. The findings have also been replicated with several different independent assays.

Plasma (or serum) biomarkers for Aβ have developed significantly over the years. The first generation of studies on plasma Aβ42 (and the Aβ42/40 ratio) showed no change in clinically diagnosed AD cases versus cognitively unimpaired elderly (for review see[25]). However, newer studies with improved assays have repeatedly shown that plasma or serum measures of Aβ42 (or Aβ42/Aβ40) are altered in AD, although typically with much lower effect size than corresponding CSF biomarkers (see e.g., refs. [54,65–69]). This may be due to that ultrasensitive Simoa immunoassay or immunoprecipitation combined with mass spectrometry have better analytical performance than the ELISA or Luminex methods used in the first generation of studies. However, a contributing factor is likely that in the older studies, a proportion of clinically diagnosed AD cases may have been misdiagnosed and some of the unimpaired elderly may have had clinically silent AD pathology, thereby limiting performance of plasma Aβ biomarkers. In contrast, more recent studies dichotomize AD patients and controls based on Aβ PET positivity and negativity (and compare Aβ PET-positive AD with PET-negative controls), which improves the possibility to find a high performance of any candidate AD biomarker.

Several proteins potentially related to synaptic or neuronal injury have been tested as biomarkers. One example is the intracellular structural protein NFL, which is increased in CSF and blood after neuronal injury, and therefore increased non-specifically in many neurological diseases (as shown for example in a large meta-analysis on 8727 patients with different diseases and 1332 healthy controls[70]). Very high NFL levels are seen in conditions with rapidly progressive neurodegenerative diseases, such as CJD (at least six studies found this, see e.g., ref. [71]) and motor neuron disease (a meta-analysis showed increased levels in 15/16 studies[72]), but levels are also increased in conditions with more slowly progressive injury, such as frontotemporal lobe dementia (a meta-analysis showed increased levels in 26/26 studies[72]), atypical parkinsonian disorders (at least four studies found this, e.g., refs. [70,73]), and vascular dementia (six

studies are summarized in a meta-analysis[74]). CSF and blood levels of NFL are also increased in AD (tested in at least 29 studies, see refs. [64,72,75–77]). Another non-specific injury marker is CSF 14-3-3, which has traditionally primarily been used in work-up of CJD[78] (a meta-analyses summarized results from 13 CJD studies[79]). Another injury-related marker, which potentially is more specific to AD, is the postsynaptic protein neurogranin[80] (increased CSF neurogranin levels in AD were seen in 10/10 studies included in a meta-analysis, using at least three different assays[64]). CSF neurogranin is also increased in MCI due to AD compared to other MCI, and has been validated for AD neuropathology using autopsy data[81]. Another example is the neuronal protein VLP-1, which has been shown to be increased in CSF and blood after brain injury[82] (increased CSF VLP-1 levels in AD were seen in 8/8 studies included in a meta-analysis[64]).

A few markers of inflammation and astroglial activation have also been reproduced. The strongest data appears to exist for YKL-40, which is a glycoprotein that can be released by several different cell-types in the body, but in the CNS it is mainly related to activated astrocytes[83]. CSF YKL-40 is increased in several neurodegenerative diseases. At least 14 studies have compared AD dementia versus controls, and all but two found increased levels in AD, but the fold change in AD versus controls in minor[64]. CSF YKL-40 was also increased in MCI due to AD compared to other MCI patients.

With regards to specific markers for non-AD pathologies, very few reproduced biomarkers have been reported, but aggregation assays for prion protein using RT-QuIC assays on CSF have high diagnostic accuracy for CJD (at least six different studies have found this, summarized in ref. [79]) and are used in clinical practice for CJD diagnosis in several countries[5–7] (e.g., US, UK, and Sweden). Similarly, RT-QuIC assays for CSF α-synuclein have been shown in several studies to have high sensitivity and specificity for PD and dementia with Lewy bodies (at least eight studies, e.g., refs. [84,85]).

**Rank II: uncertain reproducibility**. Many biomarkers for neurodegenerative diseases have either had few attempts of replication, or the evidence has been conflicting, without consensus on how to optimally use them in practice. These biomarkers may potentially be useful in the future, possibly after further standardization work. We discuss a few examples here.

One group of promising plasma-based biomarkers are P-tau with different phosphorylation variants (threonine 181 and 217)[86–90]. These correlate well with CSF P-tau and Tau PET, can differentiate AD from other neurodegenerative diseases, and predict future conversion to AD dementia. The results from the published studies on plasma P-tau are very encouraging, but we believe that the number of studies is still too small for plasma P-tau to qualify for Rank I.

sTREM-2 is released from microglia. Most studies (4/6 studies in a meta-analysis[64]) show increased CSF sTREM-2 in AD versus controls. Levels are also slightly increased in AD when subjects are enriched for AD pathology using other biomarker data[91]. Several other inflammatory markers in blood or CSF have also shown variable, but overall slight associations with AD[92]. One example is CHIT1, which has been found to be increased in CSF in both AD[93,94] and other neurodegenerative diseases[95]. These results demonstrate that although there can be a statistically significant difference in a biomarker in research studies, the degree of change can be too low to fill the demands for a useful diagnostic test in the clinic. Such biomarkers roles are mainly limited to pointing at involvement of inflammatory processes in a complex multifactorial disease. With specific treatments, for

example directed against microglia, some of these markers could also be explored as outcome measures.

CSF α-synuclein has been tested as a biomarker for PD, with varying results. In a meta-analysis of 34 studies, CSF α-synuclein was slightly reduced in PD, but the diagnostic performance was considered too poor for clinical practice, and most studies were at risk for bias[96]. Results for CSF α-synuclein in AD dementia have included both reductions, increases, or no effects in AD compared to controls[64]. One possible explanation for the poor reproducibility may be that CSF α-synuclein can be affected by multiple pathological processes. Hypothetically, the presence of Lewy bodies (a common co-pathology in AD patients) could reduce CSF α-synuclein (as in PD), while the presence of synaptic or neuronal degeneration could increase CSF α-synuclein as an injury response[97]. A pre-analytical factor affecting CSF α-synuclein is that it is sensitive to blood-contamination, which gives false high levels[98]. The inconsistent results, the unclear biological role, and the pre-analytical variability may make it difficult to apply CSF α-synuclein as a biomarker. However, we note that CSF α-synuclein may potentially be useful to separate AD from dementia with Lewy bodies[99]. There are also a few studies on α-synuclein in blood with varying results[100–103].

Plasma or serum levels of T-tau have been measured in AD in several studies, with conflicting results. Most studies have found slightly increased levels in AD (with varying effect sizes[54,104–106]), but there are also studies without difference between AD and controls[107], or with lower levels in AD[108]. One explanation for the poor reproducibility is that the results may be platform- or assay-dependent. Another possibility is that T-tau in blood may have rapid kinetics and is degraded, or is sensitive to processes beyond neurodegeneration[37]. In our opinion, this argues against using T-tau in blood as an AD biomarker.

Several other blood-based biomarkers have also shown slight associations with different AD-phenotypes (e.g., clinical diagnosis, brain Aβ burden, or atrophy measures) in several studies, for example 1-antitrypsin, α-2-macroglobulin, apolipoprotein E and complement C3[109,110]. Such reproducible associations all point to possible involvement of these proteins in the pathogenesis of AD, although the associations are generally too weak to provide clinical utility.

**Rank III: replication failure**. Many biomarkers for neurodegenerative diseases have failed replication. Here we can only discuss a few examples. In general, biomarkers discovered through exploratory methods, e.g., proteomics studies without a grounded hypothesis about links to specific disease mechanisms, have low reproducibility. One study attempted to replicate associations with AD for 94 candidate plasma proteins that had been described previously in at least one (out of a total of 21) discovery/panel-based studies (each protein had most often only been associated with AD in one previous study)[110]. Only nine proteins had significant effects in the validation cohort, meaning that they were associated with at least one of several possible AD phenotypes (including diagnosis, cognitive measures, or measures of brain structure). However, we note that it was rare for a protein to be associated with more than one AD phenotype, and some associations were even in the opposite direction compared to the original studies. Another study (on the AIBL cohort) aimed to validate blood-based proteins related to Aβ PET positivity[17]. Thirty-five proteins that had been described in at least one of four previous proteomics studies (including two AIBL studies) were tested. Only two proteins were associated with Aβ PET in the validation study. The same study also highlighted that different multiplex proteomics platforms (in this case, SOMAscan and Myriad's Rules-Based Medicine Multi-analytes Profile) may give

different results for putatively similar biomarkers. Note that panel-based biomarker discovery work still offers powerful ways to study a broad range of biological processes in neurodegenerative diseases, but findings need careful validation to avoid poor reproducibility.

Studies that build a classifier from a panel of biomarkers also have low reproducibility. One study published in 2007 used a proteomics approach for AD diagnosis[111]. Out of 120 available proteins, 18 proteins were selected (several of these were involved in immune response). Used together, the proteins had an overall accuracy of 89% for AD dementia versus controls, and successfully identified most MCI patients who later converted to AD dementia. The study included 259 individuals, recruited from seven centers. This study has been cited by over 700 papers (according to www.nature.com, October 2020). In 2012, a replication attempt was conducted in a cohort of controls, AD patients, and other patients (total $N = 433$)[112]. The performance of the tested biomarkers was considerably lower in the replication study than in the original study (AUC 0.63). For several of the tested proteins, the associations were also reversed in the second study (e.g., lower plasma levels in AD in the original study, but higher in AD in the replication study).

One study published in 2014 used a plasma lipidomics approach to detect preclinical AD[60]. A panel of ten metabolites had an impressive performance for AD diagnosis (AUC 0.92). However, in 2016 another group presented a failed attempt to replicate the performance[113]. In a subsequent debate, the authors of the original study underscored differences between the original study and the replication to explain the discrepancies. They pointed to differences in sample matrix (plasma versus serum), sample storage time (longer in the replication study), and frequency of clinical follow-up for endpoints (lower in the replication study)[114]. The authors of the replication study responded that they only found minor differences between serum and plasma, that they had included a second validation cohort to have more similar storage times as the original study, and that they considered the follow-up designs to be comparable between the studies[115]. They also underscored that the original study used a small sample for validation (21 patients and 20 controls), which they thought had overinflated the effect size. There is still no replication study published having validated the results on the diagnostic performance for AD for these ten metabolites.

### Future outlook
As the fluid biomarker field advances, we will see more novel biomarkers that are reproducible between different cohorts and methods of quantification. To guide both clinicians and policymakers it will be important to compare the value of these novel biomarkers to already established state-of the-art diagnostic methods. For example, if a new CSF biomarker for prediction of development of AD dementia is found it needs to be evaluated against already established biomarker (Aβ42/Aβ40, P-tau, T-tau), but also magnetic resonance imaging measures such as cortical thickness of the medial temporal lobe and memory function. Similarly, if a new blood-based biomarker for prediction of ALS is discovered it should be compared to plasma NFL[116].

It will also be important to show that biomarkers are reproducible not only within a highly specialized setting (tertiary or secondary referral center), but also in a primary care setting[117,118]. Poor reproducibility in primary care may be partly due to differences in pre-analytical factors, which may be more difficult to standardize in a primary care setting than in a specialized setting. Poor reproducibility could also be due to cohort-related factors, since the demographics of a primary care setting may be different than in the often highly selected population in a specialized setting (which may affect the relationship between a biomarker and an underlying pathology). For practical purposes, blood-based biomarkers will be more relevant than CSF-based biomarkers for primary care. Reproducibility for primary care will therefore require strict control of factors that can influence blood measurements. Low analytical variability will be key especially to detect subtle longitudinal changes in blood-based biomarkers for aggregation of pathologies. Despite the many obstacles and challenges in validating biomarkers in primary care, we greatly encourage this type of validation since a biomarker that passes this replication test most likely has proved to have a high level of robustness in terms of the influence from pre-analytical factors, co-morbidities and a variety of demographic factors. Such a successful validation also potentially makes the biomarker accessible for a much larger group of people.

**Summary recommendations for high-quality publications**. All stakeholders need to define what is needed in terms of validation and reporting to improve reproducibility in biomarker research (a summary of our recommendations is outlined in Figs. 1 and 2). Authors should to the best of their capacity provide comprehensive and detailed reports of their findings. Authors should aim to describe as many potential usages as possible for a novel biomarker to demonstrate convergence of findings. The most common applications are: (1) cross-sectional identification of diagnosis or another disease feature, (2) forward-looking prognostication of a feature after diagnosis is established, and (3) longitudinal biomarker measurements to track disease changes over time. A complex biomarker study may include all of these different aspects, with multiple cross-sectional comparisons and longitudinal prediction of several different variables. Naturally, an isolated positive finding together with several clearly non-significant results is a warning that the finding may be a false-positive result with low reproducibility. In contrast, multiple, convergent and logical associations for a biomarker are very encouraging. Besides testing multiple aspects of the biomarker, another hallmark of a high-quality biomarker publication is that it includes a sufficient validation section, the best being validation of results in an independent cohort of patients and controls. Another hallmark is that the biomarker is thoroughly compared with relevant state-of-the art methods. A high-quality study should also report as much data as possible on variability of the biomarker (including analytical variability and biological variability). Finally, we encourage editors to also accept well-performed negative (failed) replications (both in the same or another journal) to counteract positive publication bias. To minimize publication bias, journals may also consider accepting papers as "registered reports", where a proposed set of analyses are reviewed and can be provisionally accepted for publication before data collection has begun.

It is our hope that the recommendations in this perspective may help to improve reproducibility of future research on fluid biomarkers for neurodegenerative diseases.

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

## Acknowledgements
K.B. is supported by the Swedish Research Council (#2017-00915), the Alzheimer Drug Discovery Foundation (ADDF), USA (#RDAPB-201809-2016615), the Swedish Alzheimer Foundation (#AF-742881), Hjärnfonden, Sweden (#FO2017-0243), the Swedish state under the agreement between the Swedish government and the County Councils, the ALF-agreement (#ALFGBG-715986). Work at the authors' research center was supported by the Swedish Research Council, the Knut and Alice Wallenberg foundation, the Marianne and Marcus Wallenberg foundation, the Strategic Research Area MultiPark (Multidisciplinary Research in Parkinson's disease) at Lund University, the Swedish Alzheimer Foundation, the Swedish Brain Foundation, The Swedish Alzheimer Association, The Swedish Medical Association, The Parkinson foundation of Sweden, The Parkinson Research Foundation, the Skåne University Hospital Foundation, The Bundy Academy, The Konung Gustaf V:s och Drottning Victorias Frimurarestiftelse, and the Swedish federal government under the ALF agreement.

## Author contributions
N.M.C. drafted the manuscript. S.P., K.B., and O.H. revised the manuscript for intellectual content.

## Competing interests
N.M.C. and S.P. report no competing interest. K.B. has served as a consultant or at advisory boards for Abcam, Axon, Biogen, Lilly, MagQu, Novartis and Roche Diagnostics, and is a co-founder of Brain Biomarker Solutions in Gothenburg AB, a GU Venture-based platform company at the University of Gothenburg. O.H. has acquired research support (for the institution) from Roche, Pfizer, GE Healthcare, Biogen, AVID Radiopharmaceuticals and Euroimmun. In the past 2 years, he has received consultancy/speaker fees (paid to the institution) from Biogen and Roche.
