## [Peer Review File · Nature Communications]

Reviewers' comments:

Reviewer #1 (Remarks to the Author):

The manuscript "Increasing the reproducibility of fluid biomarker studies in neurodegenerative studies" discusses the very important topic of how to generate and validate novel biomarkers that potentially have a path to clinical use. The manuscript, however, suffers from a broad range of problems:

1. First, this manuscript very much reads as the opinion of a small select group of people. It is not a data driven review and/or analysis of different factors that impact study validity or reliability. The authors completion fail to mention any of the guidances readily available to clinical labs and the research community. Everything said in this article has been clearly defined elsewhere. Also, the authors make many statements like "we believe". This group is not a consensus panel that is reflective of the opinion of the field. If a consensus statement is intended, far more representation is required. There is nothing new being added to the field other than opinion.
2. The authors make very strong claims, that are difficult to justify. For example, the wholesale dismissal of discovery work is unfounded. There is still tremendous debate on the pathology of AD (and other neurodegenerative diseases). AD is not simply amyloid and or tau, but a host of system level dysfunction. Therefore, dismissing other markers and/or discovery work is a major mistake. One needs only to review the history from discovery to clinic of numerous biomarkers and drugs to see that serendipity is a major factor in science. The field should be encouraged to look beyond the standard existing dogma in order to better understand the disease itself.
3. The authors opinions are very broad and do not take into account biomarker use at any level. The needs for development and validation (as well as clinical application) for one biomarker use is very different from another. Blanket "to-dos" for the field are antiquated and of little use. The authors should be far more specific in what the recommendations are as relevant to specific biomarker needs.
4. The implicit bias against multi-marker approaches is also a mistake. There are IVDs built on multi-marker approaches that perform exceptionally. Additionally, outside of the AD space, many fields are trying to build multi-step and multi-marker approaches to better refine biomarkers for improved outcomes (e.g. cancer).
5. The statistical methods section is not very useful as written. Without taking into account the specific need(s) of the use of the particular biomarker, one cannot even begin to provide guidance on statistical methods.

Reviewer #2 (Remarks to the Author):

This is a review on the reproducibility of fluid biomarkers in the field of neurodegeneration from one of the main research group working in it. It provides a useful overview of the main issues and would be of general interest to neurodegeneration researchers.

My main comments are:

1. A review aiming for comprehensiveness would have benefited from a systematic approach to the summary of evidence. It is not clear to me why the authors have chosen not to do that so it may be worth justifying this decision.
2. The paper would be particular interest to Alzheimer's disease researchers. As such, it would be have been helpful to include a table with the main analytes discussed with an indication of the level of evidence as well as references for impact by each of the main factors affecting reproducibility. This will make the paper more useful to researchers designing biomarker studies in AD.
3. The authors mention newer platforms having better performance in detecting plasma AD biomarkers versus older techniques. It would appear to me that a section explaining the variability introduced by different platforms borrowing examples from AD would be apt.
4. The importance of ensuring the statistical power of validation cohort is mentioned. I think the paper

would benefit from an expansion of this point as well as a discussion of how the 80/20 ratio of training to validation was arrived at and what evidence supports its use instead of other ratios.

5. The limitations of relating biomarkers to a gold standard of syndromal diagnosis should be elaborated on further. In the case of AD, even with standard criteria such as NINCDS-ADRA the lack of agreement between clinical diagnosis and post-mortem pathology is in the region of 30%.

6. In my view the issue of test-retest reliability deserves further exploration especially in the context of research demonstrating that rapid accumulation of AD biomarkers carries information about risk for dementia that may be richer than the one based on a rigid cut-off (e.g. Landay 2018 doi: 10.1212/WNL.0000000000005354). This approach may become particularly relevant with the longitudinal follow-up of large cohorts using plasma testing. Test-retest reliability is crucial to evidence that any change is the result of accumulation versus variability in the biomarker

Minor points:

Typos: Line 39, should be on its own

P6, paragraph 2: Needs an explanation of what the difference between Ab42 and 40 is that makes AB40 better to be used as a reference

Line 393; newer instead of never?

Line 556 to instead of do?

Reviewer #3 (Remarks to the Author):

Dear colleagues,

this is an excellent review article of high clinical relevance. Most of the relevant issues addressing successful implementation of body fluid biomarkers to support the early and differential diagnosis of neurodegenerative diseases have been addressed. Moreover, the key messages are elegantly summarized in tables and figures.

However, the authors should check the manuscript again regarding semantic errors, e.g. line 39 "... on its on guide..." should be "... on its own guide..." or line 393 "However, never studies ..." should be "However, newer studies...".

Regarding, Abeta peptide ratios for the blood based early diagnosis of Alzheimer's Dementia the authors should emphasize that in contrast to CSF Abeta peptide ratios (effect size appr. 50%) the most relevant recent papers indicate an effect size of blood Abeta peptide biomarkers close or below 20%. Accordingly, only high precision Abeta peptide blood assays can be promising offering intra- and most importantly inter-assay coefficients of variations close to or below 10%. In general, in the reviewers view an additional quality criterion of a neurochemical biomarker assay is the ratio of effect size to inter-assay coefficient of variation.

Jens Wiltfang, Göttingen, Germany

Reviewers' comments:

Reviewer #1 (Remarks to the Author):

The manuscript "Increasing the reproducibility of fluid biomarker studies in neurodegenerative studies" discusses the very important topic of how to generate and validate novel biomarkers that potentially have a path to clinical use. The manuscript, however, suffers from a broad range of problems:

1. First, this manuscript very much reads as the opinion of a small select group of people. It is not a data driven review and/or analysis of different factors that impact study validity or reliability. The authors completion fail to mention any of the guidances readily available to clinical labs and the research community. Everything said in this article has been clearly defined elsewhere. Also, the authors make many statements like "we believe". This group is not a consensus panel that is reflective of the opinion of the field. If a consensus statement is intended, far more representation is required. There is nothing new being added to the field other than opinion.

- This is not an opinion paper, but a review of issues contributing to poor reproducibility of fluid biomarker studies in neurodegenerative diseases. Respectfully, we don't agree that "Everything said in this article has been clearly defined elsewhere". We are not aware of a similar up-to-date review published elsewhere. We do not think that a review paper needs to be written by an appointed consensus panel in order to be useful.

2. The authors make very strong claims, that are difficult to justify. For example, the wholesale dismissal of discovery work is unfounded. There is still tremendous debate on the pathology of AD (and other neurodegenerative diseases). AD is not simply amyloid and or tau, but a host of system level dysfunction. Therefore, dismissing other markers and/or discovery work is a major mistake. One needs only to review the history from discovery to clinic of numerous biomarkers and drugs to see that serendipity is a major factor in science. The field should be encouraged to look beyond the standard existing dogma in order to better understand the disease itself.

- The reviewer does not say what is referred to by "discovery work". Perhaps this comment from the reviewer refers to the section in the manuscript where we describe that studies testing large panels of biomarkers are liable to poor reproducibility. This should not be misunderstood as a "wholesale dismissal of discovery work", but it shows that validation is more critical when reporting biomarker panel results. We have added a sentence about this ("Note that panel-based biomarker discovery work still offers powerful ways to study a broad range of biological processes in neurodegenerative diseases, but findings need careful validation to avoid poor reproducibility", page 13, line 15ff). Nevertheless, current clinically useful biomarkers for AD were not discovered by serendipity (or "-omics") but by translating clinical neuropathology findings to fluid (and PET) biomarker. We agree that AD involves a multitude of biological processes, and biomarkers are excellent tools to study this in vivo.

3. The authors opinions are very broad and do not take into account biomarker use at any level. The needs for development and validation (as well as clinical application) for one

biomarker use is very different from another. Blanket "to-dos" for the field are antiquated and of little use. The authors should be far more specific in what the recommendations are as relevant to specific biomarker needs.

- We do not agree with the reviewer. Instead, we think that "to-dos" are very important to increase reproducibility in biomarker research, and to avoid publishing reports that fail to be reproduced.

4. The implicit bias against multi-marker approaches is also a mistake. There are IVDs built on multi-marker approaches that perform exceptionally. Additionally, outside of the AD space, many fields are trying to build multi-step and multi-marker approaches to better refine biomarkers for improved outcomes (e.g. cancer).

- The reviewer does not specify which multi-marker approaches are referred to, that are IVD approved. We are not aware of any IVD approved multi-marker tools, or clinically used multi-marker test for neurodegenerative diseases. We whole-heartedly support more research on this with carefully validated studies of multi-marker approaches. We are encouraged by the fact that other fields are also trying to explore such venues.

5. The statistical methods section is not very useful as written. Without taking into account the specific need(s) of the use of the particular biomarker, one cannot even begin to provide guidance on statistical methods.

- The statistical advice provided in this paper are general and applicable to most biomarker papers in the field.

Reviewer #2 (Remarks to the Author):

This is a review on the reproducibility of fluid biomarkers in the field of neurodegeneration from one of the main research group working in it. It provides a useful overview of the main issues and would be of general interest to neurodegeneration researchers.

My main comments are:

1. A review aiming for comprehensiveness would have benefited from a systematic approach to the summary of evidence. It is not clear to me why the authors have chosen not to do that so it may be worth justifying this decision.

- This paper was a solicited review from Nature Communications. The request from the journal was a guided analysis of key factors that contribute to poor reproducibility in fluid biomarker studies for neurodegenerative diseases (including of highly cited papers in top ranked journals). The request was not do a completely comprehensive and systematic analysis of all fluid biomarker studies in AD (which amount to several 1000s of publications). We now clarify the aim of the paper in the introduction, where we have added these lines: "This is not a systematic quantitative review of all fluid biomarker studies that have been published for neurodegenerative diseases. We have selected examples of studies that represent different types of reproducibility problems." (page 3, line 34ff)

2. The paper would be particular interest to Alzheimer's disease researchers. As such, it would

be have been helpful to include a table with the main analytes discussed with an indication of the level of evidence as well as references for impact by each of the main factors affecting reproducibility. This will make the paper more useful to researchers designing biomarker studies in AD.

- We include the main analyses and indication of the level of evidence in figure 3. We appreciate the suggestion by the reviewer to make this panel more extensive, but at the same time, we believe this would be too detailed, and too speculative since all the causes of poor reproducibility for each specific biomarker are not known in detail.

3. The authors mention newer platforms having better performance in detecting plasma AD biomarkers versus older techniques. It would appear to me that a section explaining the variability introduced by different platforms borrowing examples from AD would be apt.

- We thank the reviewer for this comment. We have now expanded this text section (page 10, lines 41ff):

” Plasma (or serum) biomarkers for A β have developed significantly over the years. The first generation of studies on plasma A β 42 (and the A β 42/40 ratio) showed no change in clinically diagnosed AD cases versus cognitively unimpaired elderly (for review see ¹). However, newer studies with improved assays have repeatedly shown that plasma or serum measures of A β 42 (or A β 42/A β 40) are altered in AD, although typically with much lower effect size than corresponding CSF biomarkers (see e.g. ^{53,64–68}). This may be due to that ultrasensitive Simoa immunoassay or immunoprecipitation combined with mass spectrometry have better analytical performance than the ELISA or Luminex methods used in the first generation of studies. However, a contributing factor is likely that in the older studies, a proportion of clinically diagnosed AD cases were mis-diagnosed and unimpaired elderly had clinically silent AD pathology, thereby limiting performance of plasma A β biomarkers. In contrast, more recent studies dichotomize AD patients and controls based on A β PET positivity and negativity (and compare A β PET positive AD with PET negative controls), which improves the possibility to find a high performance of any candidate AD biomarker.”

4. The importance of ensuring the statistical power of validation cohort is mentioned. I think the paper would benefit from an expansion of this point as well as a discussion of how the 80/20 ratio of training to validation was arrived at and what evidence supports its use instead of other ratios.

- We have added the line “The validation cohort must be sufficiently large to be powered to detect the biomarker effect found in the discovery cohort.” (page 8, line 38) There is no universal advantage of the 80/20 split compared to other fractions, and other ratios (e.g. 90/10) may be better for small sample sizes, as we now mention. We have clarified (page 8, line 47) that K-fold cross validation is preferable to a simple training/test set split, because it reduces the impact of the random grouping of subjects into one of the sets, and gives a distribution of the test effect, rather than just one test performance value.

5. The limitations of relating biomarkers to a gold standard of syndromal diagnosis should be elaborated on further. In the case of AD, even with standard criteria such as NINCDS-ADRA the lack of agreement between clinical diagnosis and post-mortem pathology is in the region of 30%.

- We are grateful for this important comment. We have added a line about this: “This is valuable since clinical diagnosis is imperfect, with about 70% sensitivity and specificity”, where we cite the large neuropathology study by Beach et al in J Neuropathol Exp Neurol (2013). (page 9, line 21ff)

6. In my view the issue of test-retest reliability deserves further exploration especially in the context of research demonstrating that rapid accumulation of AD biomarkers carries information about risk for dementia that may be richer than the one based on a rigid cut-off (e.g. Landay 2018 doi: 10.1212/WNL.0000000000005354). This approach may become particularly relevant with the longitudinal follow-up of large cohorts using plasma testing. Test-retest reliability is crucial to evidence that any change is the result of accumulation versus variability in the biomarker

- We are grateful for this suggestion, and we have added a few lines about this in the section on analytical factors, including the suggested reference to the Landau paper (“Significant variability also makes it difficult to use longitudinal rates of change as indicators of incipient pathology. Longitudinal testing of biomarkers with low variability may provide meaningful information, even before critical thresholds are reached, as shown for A β PET imaging”, page 7, line 14). We have also added a line about this in the section on primary care testing using blood-based biomarkers (“Low analytical variability will be key especially to detect subtle longitudinal changes in blood-based biomarkers for aggregation of pathologies”, page 14, line 21).

Minor points:

Typos: Line 39, should be on its own

- Thank you, we have corrected this.

P6, paragraph 2: Needs an explanation of what the difference between Ab42 and 40 is that makes AB40 better to be used as a reference

- We have added a sentence to explain this.

Line 393; newer instead of never?

- Thank you, we have corrected this.

Line 556 to instead of do?

> Thank you, we have corrected this.

Reviewer #3 (Remarks to the Author):

Dear colleagues,
this is an excellent review article of high clinical relevance. Most of the relevant issues addressing successful implementation of body fluid biomarkers to support the early and differential diagnosis of neurodegenerative diseases have been addressed. Moreover, the key

messages are elegantly summarized in tables and figures.

- We are grateful for this positive comment about our work.

However, the authors should check the manuscript again regarding semantic errors, e.g. line 39 "... on its on guide..." should be "... on its own guide..." or line 393 "However, never studies ..." should be "However, newer studies...".

- Thank you, we have corrected this.

Regarding, Abeta peptide ratios for the blood based early diagnosis of Alzheimer's Dementia the authors should emphasize that in contrast to CSF Abeta peptide ratios (effect size appr. 50%) the most relevant recent papers indicate an effect size of blood Abeta peptide biomarkers close or below 20%. Accordingly, only high precision Abeta peptide blood assays can be promising offering intra- and most importantly inter-assay coefficients of variations close to or below 10%. In general, in the reviewers view an additional quality criterion of a neurochemical biomarker assay is the ratio of effect size to inter-assay coefficient of variation.
Jens Wiltfang, Göttingen, Germany

- Thank you, we agree with this, and we now include a mentioning of the lower effect size for blood Abeta peptide biomarkers compared to CSF Abeta biomarkers ("However, newer studies with improved assays have repeatedly shown that plasma or serum measures of A β 42 (or A β 42/A β 40) are altered in AD, although typically with much lower effect size than corresponding CSF biomarkers (see e.g. ^{53,64-68})", page 10, line 44).

REVIEWERS' COMMENTS:

Reviewer #2 (Remarks to the Author):

The authors have addressed the most salient issues and I have no further comments.

Reviewer #3 (Remarks to the Author):

Dear colleagues,
I appreciate that my review suggestions have been adequately integrated into the manuscript.